# Barley systematics and taxonomy foreseen by seed morphometric variation

**Angèle Jeanty**[1]*, **Laurent Bouby**[1], **Vincent Bonhomme**[1,2], **François Balfourier**[3], **Clément Debiton**[3], **Camille Dham**[1], **Sarah Ivorra**[1], **Jérôme Ros**[1], **Allowen Evin**[1]

**1** ISEM, CNRS, EPHE, IRD, Univ Montpellier, Montpellier, France, **2** Athéna, Lacamp, Roquedur, France, **3** UCA, INRAE, GDEC, Clermont-Ferrand, France

* angelejeanty@gmail.com

**Data Availability Statement:** All raw data are fully available on a public Figshare repository [https://doi.org/10.6084/m9.figshare.21940241].

## Abstract

Since its Neolithic domestication in the Fertile Crescent, barley has spread to all continents and represents a major cereal in many modern agrarian systems. Current barley diversity includes thousands of varieties divided into four main categories corresponding to 2-row and 6-row subspecies and naked and hulled types, each of them with winter and spring varieties. This diversity is associated to different uses and allow cultivation in diverse environments. We used a large dataset of 58 varieties of French origin, (1) to assess the taxonomic signal in barley grain measurements comparing 2-row and 6-row subspecies, and naked and hulled types; (2) to test the impact of the sowing period and interannual variation on the grains size and shape; (3) to investigate the existence of morphological differences between winter and spring types; and finally (4) to contrast the relationship between the morphometric and genetic proximity. Size and shape of 1980 modern barley caryopses were quantified through elliptic Fourier Transforms and traditional size measurements. Our results indicate that barley grains record morphological diversity of the ear (89.3% classification accuracy between 2-row/6-row subspecies; 85.2% between hulled and naked type), sowing time of the grains (from 65.6% to 73.3% within barley groups), and environmental conditions during its cultivation and varietal diversity. This study opens perspectives for studying archaeological barley seeds and tracing the barley diversity and evolution since the Neolithic.

## Introduction

Barley (*Hordeum vulgare* L.) is one of the most important cereal crops in the world with a global production of 160 million tons every year (http://www.fao.org/). Barley tolerates drier, colder and poorer soils than wheat, which explains its wider geographical distribution in Eurasia [1, 2]. Today, barley grains are mainly used for both animal and human consumption while straw is dedicated to livestock, and starch is used in food production and chemical industry [3–5]. At the global scale, several hundreds of barley varieties are recorded today [6].

Two *Hordeum* subspecies can be distinguished based on spike morphology and number of fertile spikelets present at each node of the rachis. Six-row barley (*Hordeum vulgare* subsp. *vulgare* also called *Hordeum hexastichum*) has 3 fertile spikelets while in two-row barley

**Funding:** This work was supported by the European Research Council (ERC) under the European Union's Horizon 2020 research and innovation programme (grant agreement No. 852573). https://erc.europa.eu/homepage The funders had no role in study design, data collection and analysis, decision to publish, or preparation of the manuscript.

**Competing interests:** The authors have declared that no competing interests exist.

(*Hordeum vulgare* subsp. *distichum*) only the central spikelet is fertile [1, 2]. Within these two subspecies, hulled and naked types are distinguished on the basis of the adherence or non-adherence of the protective envelopes of the caryopses. Hulled barley is the most widely grown type, mainly for animal feeding and malt production for brewing. Naked barley is more scarcely cultivated nowadays and mainly serves as a human food source [1, 4, 5, 7]. In addition, barley varieties can be divided into winter- or spring-sown varieties [1, 4] that differ mainly in their vernalization requirements [8]. Winter barley has a cycle of about 10 months. Sown in autumn, it needs vernalisation, *i.e.* period of low temperatures, for its development and closes its cycle before the summer droughts. The need of vernalization ensures that ear development takes place after the risk of frost damage has passed [9, 10]. In spring barleys, flowering is not inhibited (no vernalization) because this stage takes place during the good season [11]. They have a short growth cycle, and are well adapted to northern regions where winter conditions are too harsh for the cultivation of winter varieties. Spring barleys, however, usually provide lower yields than winter barleys.

Barley diversity has been partially studied using molecular markers, and the genotyping of 570 French accessions (784 SNPs spread on the whole genome) shows that the main variation is caused by the difference between the sowing season (winter *vs*. spring), prior to the number of spike rows (6 *vs*. 2) [12].

Current barley diversity reflects its past history, which is, at least partially, known through the study of archaeobotanical remains as well as recent genetic studies. Barley is one of the founder crops of the Old World food production [2]. It was domesticated from its wild progenitor *Hordeum vulgare* subsp. *spontaneum* around 10.000 years ago in the Fertile Crescent. Only the central spikelet of this ancestor is fertile and produce a kernel. Barley domestication has long been thought to be monophyletic [13], but recent genetic analyses of current varieties support the hypothesis of a polyphyletic and multiregional origin of barley domestication [14–17] and discerned European and Asian routes of barley spread based on Simple Sequence Repeat (SSR) markers [18, 19]. Nevertheless, the biogeographical history of barley can be traced mainly from the macro-remains (grain, spikelet or chaff) found in archaeological sites [2]. They give evidence of the spread of barley in Europe and Asia as early as the Neolithic. Palaeogenetic analyses gave some insights into the early domestication and spread of barley [18, 20, 21]. However, most of the cereal macro-remains are, especially in the Mediterranean area, preserved by charring which is detrimental to ancient DNA preservation and palaeogenetic studies [22–24]. Tracing the ancient history of barley from the study of macro-remains must therefore primarily rely on morphological characteristics. The ratio of straight and twisted caryopses has long been used to identify the presence of two-row and six-row barley. In addition, the presence of naked or hulled barley types can be inferred from the general aspect of the grain and of its outer surface, naked barley grains being more roundish, especially in cross-section, and having cross-ripples on their surface [e.g. 25]. In the Near-East, morphological traits allow to document the presence of six-row hulled and naked barleys soon after the domestication of the two-row morphotype, which is closer to the wild *H. vulgare* subsp. *spontaneum* [2]. However, it is difficult to apply systematically and consistently between studies using qualitative morphological criteria. As a consequence, important issues in the history of barley spread and in the cultivation dynamics of barley types remain unclear or under debate. *Hordeum* types are not always distinguished in archaeobotanical studies. However, in Western Europe, it is often considered that the Early Neolithic agriculture relied mostly on naked six-row barley [e.g. 26, 27]. Two-row barley is however occasionally identified, in particular by chaff, and hulled six-row barley seems to be predominant in Italy and North-Eastern Spain [28, 29].

For these reasons, it is crucial to investigate quantitative (and therefore more objective) methods for discriminating and identifying barley types from seed morphology.

Several studies have shown the interest of morphometrics to discriminate barley types. Traditional morphometrics, based on the study of the length, width and thickness of archaeological barley grains, has been used to confirm the presence of 2-row barley during Roman times in France [30]. More recently, geometric morphometric analysis of grain's outline shape of a sample of 10 present-day varieties evidenced that 2-row and 6-row types could be differentiated even when the grains were experimentally charred [31]. This discrimination was later confirmed by [32] who also showed that within 2-row and 6-row barley, a selection of British and Scandinavian varieties could be distinguished. This last study also gave evidence that differences between barley types are independent of environmental conditions. In addition, it has been suggested that barley size was related to culinary systems and traditions in prehistoric Asia [33].

The north-western Mediterranean basin host an important barley diversity and France is the 5th largest producer of barley in the world according to FAO data available in 2020, with hundreds of very diverse varieties recorded [34, 35], providing a favourable context to investigate morphometric variation within and among barley types.

Based on the quantification of the size and shape variation of barley grains belonging to a set of 58 French varieties, the present study aims to (1) assess more globally how the morphological variability of caryopses is structured according to the different categories of barley (2-row vs six-row, hulled vs naked, varieties), (2) take into account the impact of inter-annual variation, (3) investigate the existence of morphological differences between winter and spring types, and (4) explore the relationship between morphometric and genomic proximities.

## Material and methods

### Morphometric data

A total of 1980 barley seeds corresponding to 58 varieties were studied (S1 Table). The varieties are divided into four taxonomical groups (Table 1): two-row naked (N = 13), two-row hulled (N = 23), six-row naked (N = 10) and six-row hulled (N = 12). The varieties are also divided into three different sowing periods: spring (N = 24), winter (N = 31), and alternative (N = 3) without seasonal preference (Table 1). So-called alternative varieties corresponding to varieties that can be sown in spring or winter.

An accession includes grains from a single variety grown and sampled the same year in the same field. A total of 66 accessions were analysed. This includes 54 varieties collected a single year and 4 varieties repeatedly harvested on three different years (varieties 10004-CFL33 and 10024-ESV are 6-row-hulled-winter barley; varieties 12510-DLG and 12900-CHI are 2-row-hulled-spring barley). These last 12 accessions (4 varieties x 3 years sown) were used to test inter-annual grain size and shape variation.

**Table 1. Number of varieties studied per taxonomic and systematic groups.**

|  | Spring | Winter | Alternative |
|---|---|---|---|
| 2-row hulled | 17 | 6 | - |
| 2-row naked | 7 | 6 | - |
| 6-row hulled | - | 12 | - |
| 6-row naked | - | 7 | 3 |

All the seeds originated from the Biological Resources Centre (BRC) small grain cereals at INRAE in Clermont Ferrand (France) were they were cultivated under the similar growing conditions and stored under strictly controlled conditions.

Once received at the laboratory, the grains were placed for 48 hours in a freezer, followed by 24 hours at 38°C in an oven in order to avoid germination and eliminate pests. Each grain was then manually peeled to remove the husks. Only complete and undeformed grains were selected.

Because the analysis of samples of 50 grains from two varieties revealed that a sample size of 30 grains was sufficient to capture the size and shape variation of a population, 30 barley seeds were analysed per accession. This preliminary analysis was performed on variance estimation using rarefaction curves [36–38] (S2 Fig).

The grains were positioned on plasticine and photographed in their dorsal and lateral views using Olympus SZ-ET microscope and DP26 Olympus Camera. The lateral view documents the shape of the grains taken in their thickness while the ventral view documents the shape of the grains visualised with the furrow facing upwards. The combination of the two views allows to document the shape of the grains from two different points of view, which somehow allows to study the grains in 2.5D, approaching a 3D description. A centimetric scale was included in all pictures. The background of each picture was removed, and the grain was converted to a black mask using Adobe Photoshop CS6 software. The 2D outline coordinates was then extracted using R v. 4.1.1 and the package Momocs v. 1.3.0 (https://github.com/MomX/Momocs/; [39]). The outline coordinates were scaled using two landmark coordinates manually localized 1 cm apart on the original picture using Image J [40]. The length, width, thickness and centroid size of the grains were calculated using the Momocs package (S1 Fig).

## Genetic data

Genetic data were available for 51 of our varieties [12], corresponding to 784 SNPs markers covering the 7 chromosomes of the genome, and were used to compute Sokal and Michener genetic distances [41] between varieties. The genetic distance matrix was compared to the morphometric distance matrix using a Mantel test.

## Statistical analysis

Differences between groups were first explored for size, shape and form (size + shape) separately. Different levels of investigation were explored: between accessions, varieties, systematic and taxonomical groups, between sowing period, and between year of collect for 4 varieties.

For size analysis, differences in length, width, thickness and centroid size of the grains were tested using Kruskall-Wallis rank tests and visualised with boxplots. Pairwise differences were tested using Wilcoxon rank tests.

To analyse shape, outlines coordinates were centred and scaled. Subsequently, the elliptical Fourier transforms (EFT) were calculated. Outline of the grains are decomposed into a series of coefficients of trigonometric functions, the harmonics. The shape of the studied object is reconstructed using the inverse transform. In our case, the lateral view of the barley grain is described by 5 harmonics and the ventral view by 7. The number of harmonics was determined using the harmonic power criterium in the Momocs package [39]. These harmonic coefficients correspond to shape variables and are analysed first using a Principal Component Analysis (PCA) in order to reduce the dimensionality of the data, assess the overall grain shape variation and detect potential outliers. Then, differences between groups were tested using Multivariate Analyses of Variance (MANOVA). Pairwise differences were assessed using pairwise multilevel comparison (vegan & pairwiseAdonis R packages). Subsequently, discriminant

analyses were used to separate groups and to accuracies presented were calculated using leave-one-out cross-validation (CVP) and their confidence intervals (MASS R package, [42]). Neighbor Joining (NJ) dissimilarity networks, based on the Mahalanobis distances, were computed. Differences in mean shapes were visualised using the MSHAPES function of the Momocs R package.

## Results

### Overall morphometric variation between varieties

Overall, barley varieties differ in their grain size (length, width, thickness, centroid sizes), shape and form of both their lateral and ventral views (all p < 2.2e-16). The centroid sizes of the ventral and lateral views appeared highly correlated with each other and with grain length (Fig 1A). Consequently, centroid sizes were not analysed further. Conversely, length, width and thickness show significative (all p-value < 2.2e-16) small correlation (Fig 1A) and were analysed separately.

Visualisation of between-varieties size variation (Fig 1B–1D) shows a strong general overlap with however some varieties showing larger or smaller values, neither appear related to the 2-row or 6-row, not to hulled or naked categories.

The between-varieties dissimilarity networks for the lateral and ventral shapes (respectively Fig 1E and 1F) appeared, at least partially, taxonomically structured. The ventral shape network (Fig 1F) is mainly structured by a 2-row vs 6-row varieties opposition with only few varieties not clustering within their categories, while the lateral shape network (Fig 1E) appeared less structured. When the lateral and ventral shapes were combined (Fig 1G), the network showed a clearer pattern, with a clear structuring, first between the 2-row and 6-row varieties,

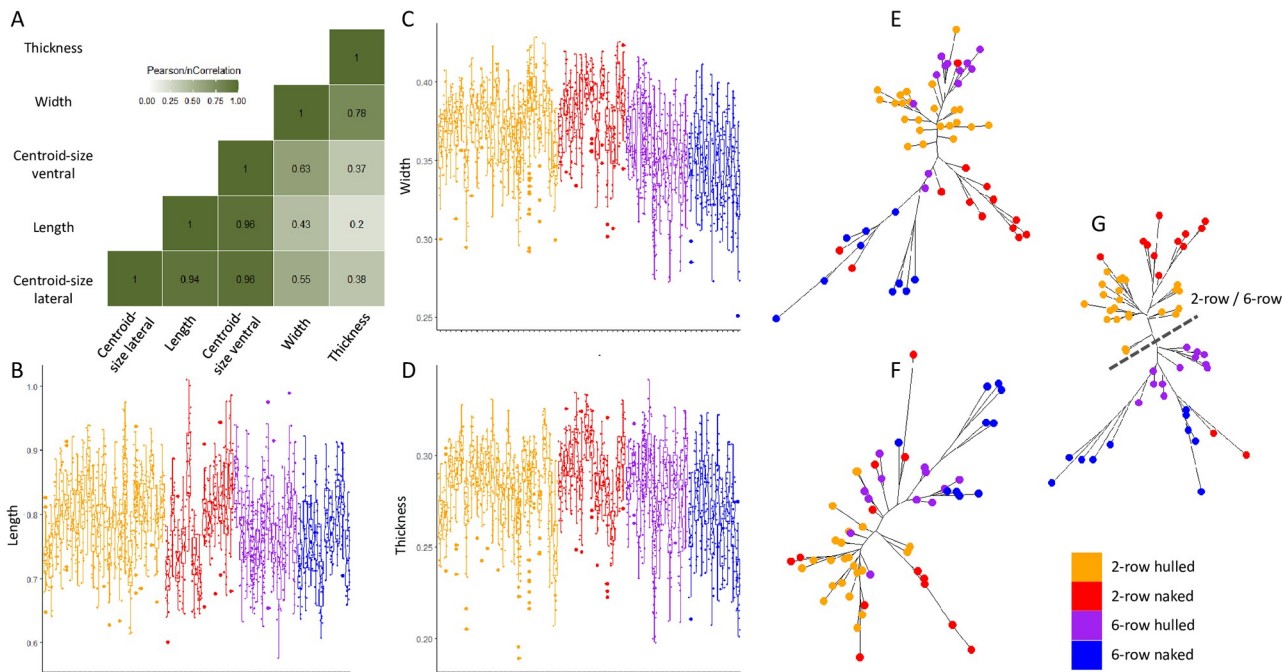

**Fig 1. Overall morphometric variation between varieties.** Correlation between the size indices of length, width, thickness and centroid sizes (CS) of the ventral (VV) and lateral (VL) views of the grains (A). Boxplot of length(B) width (C) and thickness (D) showing variation between varieties. Dissimilarity network between varieties for the lateral (E) and ventral (F) views of the grain, and the shape of the two views combined (G). The colors differenciate the hulled (red) and naked (orange) 2-row varieties and hulled (purple) and naked (blue) 6-row varieties.

**Table 2. Interannual variation in size and shape for four varieties.**

| | Length | | Width | | Thickness | | Lateral shape | | Ventral shape | |
|---|---|---|---|---|---|---|---|---|---|---|
| Variety | $\chi^2$ | p | $\chi^2$ | p | $\chi^2$ | p | F | p | F | p |
| 10004-CFL33 | 2.27 | 0.32 | 5.85 | 0.054 | 2.26 | 0.32 | 2.49 | 0.041 | 1.64 | 0.114 |
| 10024-ESV | 27.94 | 8.6e-07 | 21.7 | 1.9e-05 | 33.72 | 4.78e-08 | 5.85 | 0.001 | 7.36 | 0.001 |
| 12510-DLG | 4.45 | 0.11 | 6.68 | 0.035 | 11.48 | 0.0032 | 5.99 | 0.001 | 6.004 | 0.001 |
| 12900-CHI | 13.1 | 0.001 | 9.08 | 0.011 | 8.48 | 0.014 | 10.09 | 0.001 | 13.5 | 0.001 |

Results of Kruskal Wallis tests for size ($X^2$ and p-value) and MANOVA for shape (F and p-values). P-values are still significant after adjustment for multiple comparisons.

then between naked and hulled varieties within the two main groups. Only three exceptions can be noted corresponding to three 2-row varieties being clustered with the 6-row varieties.

## Interannual variability

Four varieties were sampled for three different years, but not necessarily in the same years, which limits direct comparisons. The varieties show different pattern of interannual size variation with one variety (10004-CFL33) showing no variation, two varieties (10024-ESV and 12900-CHI) showing interannual differences for all comparisons, and one (12510-DLG) showing significant differences only for grain thickness (Table 2 and Fig 2A–2C). The boxplots and dissimilarity network evidenced the interannual variation in grain size (Fig 2A–2C) and shape (Fig 2D, both views combined) to be weaker than variation between varieties.

## Genetic information and grain morphometrics

Genetic and morphometric datasets revealed no correlation whether it be for the lateral (p = 0.889) and dorsal (p = 0.326) shape of the grains or their length, width, and thickness (p > 0.005).

## Morphometric differences between systematic and taxonomic groups

All pairwise comparisons of the size indices between the four main categories appeared significant (all p < 2.2e-16), except length difference between 6-row hulled and naked types, and thickness difference between the 2-row and 6-row hulled types (Table 3, Fig 3A–3C). Size measurements greatly varied between sowing seasons with hulled spring varieties showing larger measurements than winter varieties when the differences were significant (Fig 3A–3C) and winter varieties showing larger measurements than winter varieties in naked types. For 6-row naked, winter varieties showed larger measurements than varieties characterized as "alternative".

Regarding dissimilarity networks, the same patterns were observed with the naked vs hulled dichotomy for the lateral shape (Fig 3D), and the 2-row vs 6-row dichotomy for the ventral shape (Fig 3E). The two sowing periods of the same category clustered together in both networks, with the 6-row (both naked and hulled) categories showing more differences between sowing periods than their 2-row counterparts (Fig 3D and 3E). The network pooling the shape of the two views (not shown) was highly similar to those based on the ventral shape.

## Discriminating power of the different morphometric parameters

All varieties were then grouped, first according to the four initial categories (2-row/6-row, hulled/naked) to which was then added the sowing period.

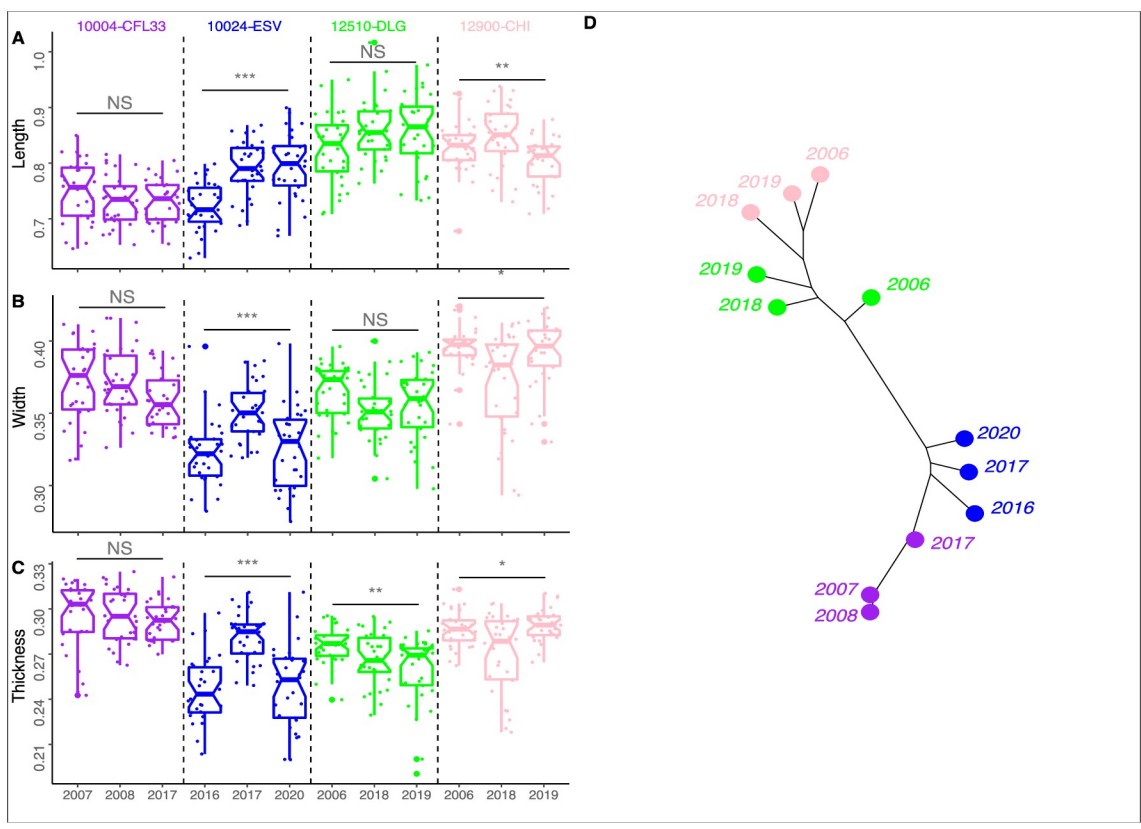

**Fig 2. Interannual variation in grain morphometrics.** Boxplot of the length (A), width (B) and thickness (C) of the grain for visualising size differences between the sampled years for four varieties. Dissimilarity network based on shape (two views combined) variation between years for four varieties. Dissimilarity network based on shape (two views combined) between the four varieties and their three sampled years (D).

For all comparisons of the four main categories (Fig 4, Table 4), shape and form performed equally, and the mean CVP obtained when the lateral and ventral views of the grains are combined always performed better than their separate analyses for comparing the taxonomic groups (Fig 4A). Results are more contrasted regarding the sowing season for which the ventral view performs equally or better than the combined analyses (Fig 4B). The three size indices provided always much lower CVP than shape and form analyses.

At best, grains are identified to the correct variety with a mean CVP of 48.1% (Confidence Interval (CI): 47.5–48.6%, shape of both views combined).

The ventral shape performed better in discriminating the 2-row vs 6-row types, while the lateral shape performed better for hulled vs naked discrimination. Visualisation of mean shape differences (Fig 5) revealed shorter and wider grains for naked compared to hulled barley in

**Table 3. Pairwise comparisons between taxa in grain length, width and thickness.**

|  | 2-row hulled | | | 2-row naked | | | 6-row hulled | | |
|---|---|---|---|---|---|---|---|---|---|
|  | Length | Width | Thickness | Length | Width | Thickness | Length | Width | Thickness |
| 2-row naked | 1.4e-05 | 8.6e-10 | 6.2e-11 | - | - | - | - | - | - |
| 6-row hulled | < 2e-16 | < 2e-16 | 0.87 | 0.014 | < 2e-16 | 1.5e-08 | - | - | - |
| 6-row naked | 2.7e-15 | < 2e-16 | < 2e-16 | 0.0098 | < 2e-16 | < 2e-16 | 0.549 | 0.0023 | 2.6e-12 |

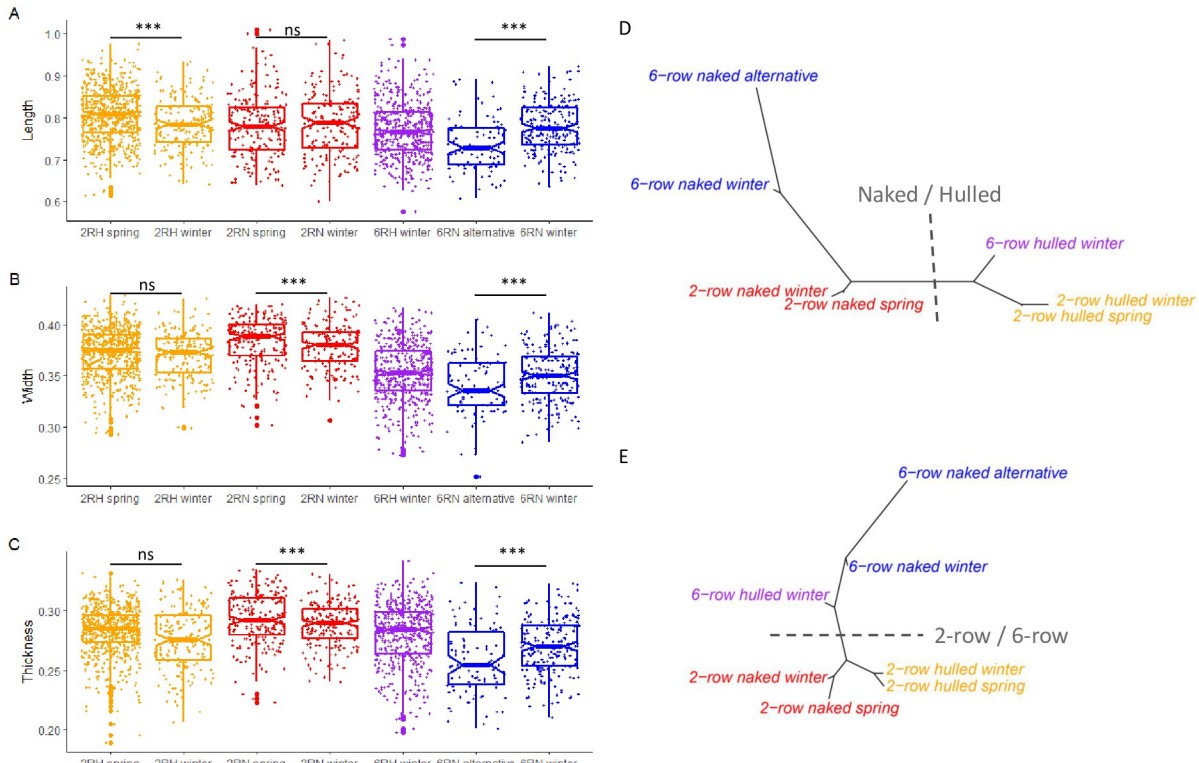

**Fig 3. Overall morphometric variation between systematic and taxonomic categories.** Boxplot of length (A) width (B) and thickness (C) showing variation between categories. Dissimilarity network between categories for the lateral (D) and ventral (E) views of the grain. The colors differentiate hulled (h, orange) and naked (n, red) 2-row varieties and hulled (h, purple) and naked (n, blue) 6-row varieties.

ventral view, but also an apex mismatch in lateral view. The differences between 2-row and 6-row barley are mainly in the lower part of the grain, with 2-row barley being shorter, wider and rounder than 6-row barley in ventral view. In the lateral view, there is also a shift in the apex of the grain and the 2-row barleys are more rounded in the furrow of the grain.

The grains can be attributed to the 2-row or 6-row category with a mean CVP of 89.3% (CI: 88.8–89.9%, form of both views combined) and to the naked or hulled category with a mean CVP of 85.3 (CI: 84.6–86.1%, shape of both views combined).

When the four categories are considered simultaneously, the lateral and ventral views performed equally, less efficiently than their combined analysis that allow attributing the grain correctly with a mean CVP of 76.7% (CI: 75.7–77.6%, shape of both views combined).

Within each of the four main categories, the sowing period can be identified with relatively high CVP ranging from 65.1% (CI: 62.2–68.6%, shape of both views combined) for two-row hulled barleys to 77.6% (CI: 73.3–81.1%, shape of both views combined) for six-row naked barleys. The sowing season of hulled barleys was more discriminated than that of naked barleys. While both views performed relatively equally for hulled barleys, the ventral view performed better for naked barleys for discriminating the sowing seasons.

## Spring, winter or alternative barley

For each of the four groups, spring varieties have longer, wider and thicker grains than winter varieties (Table 5, Fig 3). The only exception is 2-row naked, showing no length differences between spring and winter varieties.

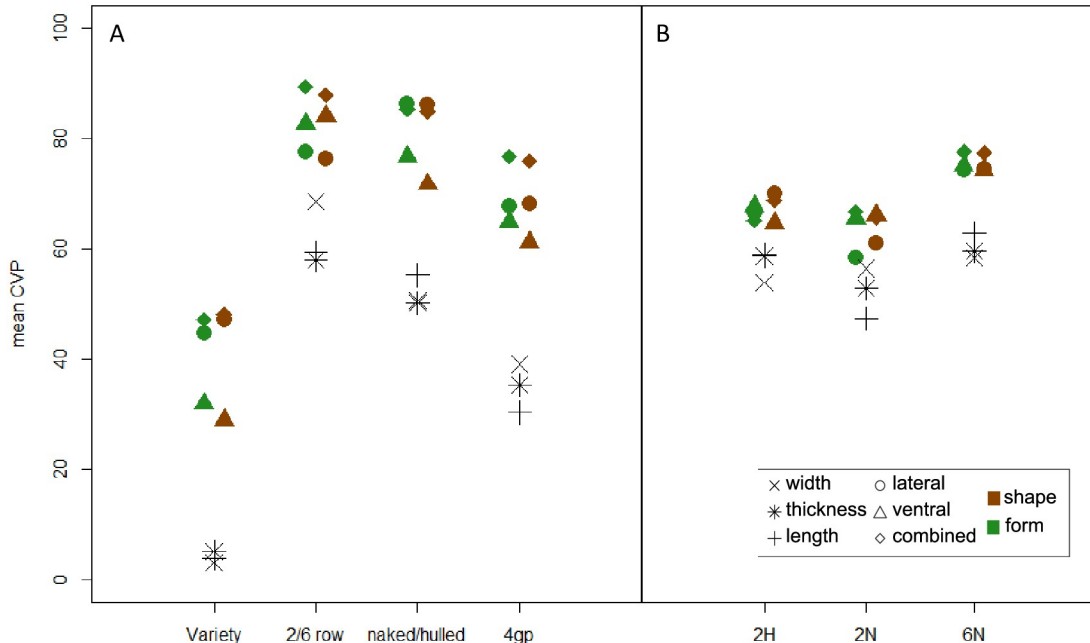

**Fig 4. Systematic and taxonomic signal in grain morphometrics.** Mean Cross-Validation Percentages (CVP) computed from width, thickness and length size indices, as well as from lateral and ventral shape and form of the grain analysed separately and combined contrasting the categories mentioned along the x-axis (A). The '4groups' (4gp) comparison includes the 2-row hulled (2H) and naked (2N), and 6-row naked (6N) categories. The sowing periods were contrasted for each category separately comparing winter vs. spring varieties for the 2H and 2N, and winter vs. alternative for the 6N (B). All mean CVP values and their confidence intervals can be found in (Table 5).

## Discussion

### Morphometric methods

Geometric morphometric analysis of barley grains belonging to 58 French varieties show that 6-row/2-row, hulled/naked types and even varieties differ in their grain size, shape and form. As expected, form and shape variables (lateral, ventral outlines and the combination of the two) are more discriminant than size variables (length, width, thickness, lateral and ventral centroid size).

Barley varieties differ in the size, shape and form of the grains. At best, 48.1% of the grains are correctly re-assigned to the correct variety using a combination of lateral and ventral

**Table 4. Differences between the 4 groups of barley in size and shape between.**

|  | $\chi^2$ | p | CVP |
|---|---|---|---|
| Length | 105.19 | < 2.2e-16 | 30.3% (CI: 29.2–31.9%) |
| Width | 373.05 | < 2.2e-16 | 39.0% (CI: 37.9–40.3%) |
| Thickness | 155.78 | < 2.2e-16 | 35.3% (CI: 34.3–36.4%) |
| Lateral shape | 51.328 | < 2.2e-16 | 67.8% (CI: 66.3–69.2%) |
| Ventral shape | 33.169 | < 2.2e-16 | 64.8% (CI: 63.3–66.0%) |
| Lat. & vent. shapes | 36.185 | < 2.2e-16 | 76.7% (CI: 75.7–77.6%) |

Results of the t-test of size, MANOVA for shape. Cross-Validation Percentages (CVP) with confidence intervals (CI). The CVP and CI are reported in Fig 5.

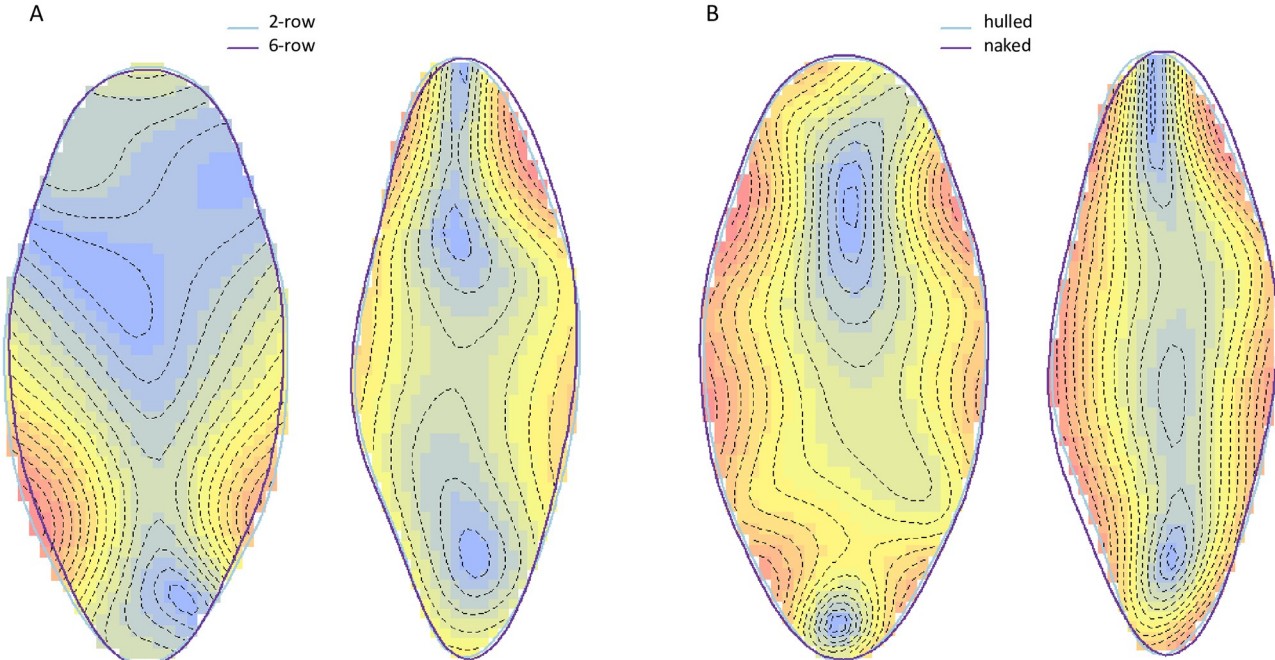

**Fig 5. Meanshape differences between hulled and naked varieties (A) and between 2-row and 6-row varieties (B).** Outlines indicate mean shapes for each of barley types: hulled (A) and 2-row (B) in blue, naked (A) and 6-row (B) in purple. Warmer colours indicate the largest morphological differences between barley features.

shapes. Our results are coherent to previous studies, [31] obtained 53.7% of correct cross-classification but using only 10 varieties and a different methodological approach based on sliding semi-landmarks on the ventral view of the grains. In 2019, [32] obtained 61.5% of correct classification based on the study of 54 landraces using an EFT analysis of the dorsal outline only. It appeared therefore possible that outline analysis, using the EFT method, better discriminate barley varieties than Procrustes approaches, though a quantitative comparison of the methods is still lacking.

The combination of the lateral and ventral grain outlines also provides a clear accuracy gain to analyses based on a single view. Here, the combination of lateral and ventral shapes increases the correct reclassification rate by about 5% compared to lateral shape alone. On the other hand, the use of a size variable in combination with lateral and ventral shapes (form) does not significantly improve the performance of the model. The better performance of the analyses based on the combination of lateral and ventral shapes is confirmed at all levels (6-rows/2-rows, hulled/naked).

**Table 5. Taxonomic variation in size and shape for each of the 4 groups of barley.**

|  | Length | | | Width | | | Thickness | | | Lateral shape | | | Ventral shape | | | Shape of the 2 views |
|---|---|---|---|---|---|---|---|---|---|---|---|---|---|---|---|---|
|  | $\chi^2$ | p | CVP | $\chi^2$ | p | CVP | $\chi^2$ | p | CVP | F | p | CVP | F | p | CVP | CVP |
| 2-row hulled | 15.48 | 8e-05 | 58.9% | 3.63 | 0.056 | 53.9% | 12.38 | 4e-4 | 58.7% | 7.20 | <2.2e-16 | 66.6% | 6.85 | <2.2e-16 | 67.7% | 65.1% |
| 2-row naked | 0.20 | 0.65 | 47.3% | 8.34 | 0.004 | 56.4% | 5.89 | 0.015 | 52.8% | 2.27 | 0.002 | 58.3% | 4.31 | 2e-11 | 65.4% | 66.7% |
| 6-row naked | 29.66 | 5e-08 | 62.8% | 9.57 | 0.002 | 58.3% | 12.88 | 3e-4 | 59.5% | 6.07 | 2e-13 | 74.2% | 7.95 | <2.2e-16 | 74.9% | 77.6% |

Results of Kruskal Wallis tests for size ($\chi^2$, p-value and CVP) and MANOVA for shape (F, p-values and CVP).

A possible explanation for the non-correlation between genetics and morphometrics may be the low number of SNPs used. To find grain measurements causative SNPs, a Genome-Wide Association Study (GWAS) would have to be performed with hundreds of thousands of SNPs [e.g. 43].

## Inter-annual variation and influence of environmental conditions in grain morphology

Our results show that grain size, shape and form vary from one year to another. This interannual variability appeared specific to each variety and no common variation emerged. That being said, interannual variation in grain shape is here shown to be lower than varietal variation. This conclusion echoes with the findings of [32]. In their study, 54 accessions were sown in two different environments and they show that the effect of growing conditions and grain morphology was minor compared to genetic factors. Evidencing that grain morphology is more strongly determined by genetics than by environmental conditions is an important milestone with the prospect of applying these models established on modern accessions to identify barley types to which belong archaeological grains. The role of genetics in the control of grain morphology is supported by the identification of QTL involved in grain form determination [44, 45] that do not seem associated with environmental variability.

Environmental conditions interact with genetics to influence barley grain size and weight [46]. The environmental effects vary according to seasonality and barley cycle [47]. The conditions during pre-anthesis (flowering) period directly influence grain weight and size through determining assimilates, while conditions during post-anthesis period influence cellular division, grain filling or deposits of starch grains. Grain width and thickness are controlled by cell division and grain length by the elongation within the developing endosperm, a process that ceases 20–25 days after initiation of flowering [47]. According to [48], barley grain length is weakly affected by the environment. On the other hand, width and thickness vary more strongly according to environmental factors [47, 49]. Our results showed differences in length, width and thickness for some varieties sown in several different years.

## Patterns of grain morphological variability in relation to spike anatomy

More broadly, the dissimilarity networks calculated on the lateral and ventral grain shapes combined indicates that our dataset is firstly structured by the number of spike rows (2-rows vs. 6 rows), then according to hulled/naked types. The discrimination of 6-row/2-row varieties is mainly determined by the ventral shape while the discrimination of hulled/naked varieties is determined by the lateral outline. This structuration is readily explainable by the constraints imposed by the anatomy of the spike on the morphology of the grains. These constraints are very different for the 6 and 2-row types. In 2-row barley only the central spikelet is fertile at each node of the rachis whereas the 6-row barley has three fertile spikelets at each node. This difference in the number of fertile grains attached to a rachis node induces a twisting of the lateral grains in the 6-row type, which will have less place to develop in the ear and thus will have a different shape from the untwisted central grains [31, 48]. A mutation on the vrs1 gene of the chromosome 2 is responsible for these differences in the fertility of lateral spikelets [50–52]. Morphological differences between hulled and naked barley grains are related to husks adherence to the grain after ripening, tight in hulled barleys and much looser in naked barley varieties [31, 48]. These differences, here quantified and visualized, are well known to archaeobotanists and especially the fact that naked barley grains are rounder than grains from hulled varieties [25]. Here we can see that the naked grains are more particularly rounded on the lateral view.

Cross-validation percentages are coherent with the observed structuration, with the accuracy for discriminating 2-row from 6-row barley being higher for the ventral shape, whereas the CVP for discriminating naked from hulled barley is higher for the lateral shape. Additionally, the combination of lateral and ventral views allows a slightly better discrimination of 6-row/2-row types than hulled/naked types. Grains can be attributed to 2-row or 6-row categories with a mean CVP of 89.3%, and to naked or hulled categories with a CVP of 85.3%, and, combining the four categories, to a CVP of 76.7%. The CVP values for 6-row/2-row types are close to those obtained in previous studies by [31] (CVP = 91%) and [32] (CVP = 87.6%). In their study, [32] only included the central grains of 6-row varieties to compare them to 2-row grains. The strong discrimination they obtained show that even the morphology of central, "untwisted", grains is constrained by the pressure of lateral spikelets in 6-row varieties. Here we decided to consider the whole population of 6-row grains, "twisted" and "untwisted", in order to develop approaches that can be used to identify archaeological barley grains which cannot always be easily sorted according to "twisted" and "untwisted" categories. In this regard it is important to note that geometric morphometrics gives encouraging results to distinguish the four types 6-row-hulled, 6-row-naked, 2-row-hulled and 2-row-naked when uncharred grains are studied.

## Morphological variation and sowing period

Contrary to genotypic data, which evidence a structuration primarily dependent on the sowing period [12], grain morphometric variation appeared less structured between winter and spring varieties. This is consistent with the fact that spike anatomy does not impose different morphological constraints between spring barley and winter barley, unlike the differences between 6-row/2-row and hulled/naked types. A weak morphological structuration nevertheless exists and is strong enough to identify winter/spring types when looking at the combined lateral-ventral shapes of the grain. Within each of the 6-row/2-row vs hulled/naked groups, relatively high CVPs, from 65.1% to 77.6%, are obtained for the distinction between winter/spring types.

Several hypotheses could be suggested to explain morphological differences between winter/spring types, since size and shape differences could be linked to genetic, physiologic or environmental influence during life cycle.

A first hypothesis could be genetic with the association between grain form QTLs [11, 44, 45, 53, 54] and loci related to phenological traits. Flowering time genes are classified into at least three families [55]: photoperiod genes (e.g. Ppd-H1), vernalization genes (e.g. Vrn-H1, Vrn-H2 and Vrn-H3, sgh1, sgh2, sgh3) and earliness per se (eps) genes, the last controlling flowering independently from photoperiod and temperature (e.g. Sdw1 for semi-dwarfing genes). Barley size QTLs are reported to be associated to Ppd-H1 locus [47, 56], eps2 locus [47, 56, 57], swd1 locus, which are linked to late maturity, reduced plant height, increased tillers number and biomass production [47].

The second hypothesis to explain the differences between spring and winter barley grains morphology could be ecological and linked to resources trade-offs. The plant has limited resources for its growth until maturity, which necessarily induces trade-offs in the allocation of these resources to the different plant sinks. For example, tillers formation could limit the resources available for grain filling, which could limit the size of the grains even though their number per plant would be greater. Tillering is the production of multiple stems (tillers) starting from the initial single seedling. This ensures the formation of dense tufts and multiple ears [8]. Tillering is influenced by genetic variation [8, 58–60] but also by environmental variation during pre-anthesis phase [58, 61, 62]. Tillering should be higher for winter barley than spring barley according to [8]. Thus, spring varieties that have less tillers than winter varieties could

provide more resources for seed filling, which would explain why spring grains are wider and thicker than winter ones.

Another explanation to differences between spring and winter barley can be variation in environmental conditions during barley growth cycle. Several critical growth periods, as tillering, flowering, filling and ripening, are sensitive to rainfall [63], drought [64, 65], available soil water [66]. The impact of environmental variability could be increased in our dataset as in the BRC of Clermont-Ferrand varieties were not always sown in the season they were supposed to be, i.e. spring varieties were occasionally sown in December because droughts during the spring sowing period are more frequent and winters are less harsh due to climate change. Unfortunately, this data was not always recorded, it would therefore be important to further investigate this question based on a detailed record of sowing time and environmental conditions along the life-cycle, and to grow the same accession at different periods.

## Conclusion

This study of 1980 grains from 58 modern varieties demonstrated the possibility of determining barley characteristics (2 rows/6 rows, naked/hulled, spring/winter) using morphometric analysis of caryopses. Despite inter-annual variability, the characteristics related to varietal differences allow varieties to be distinguished with a cross-validation percentage of 48.1%. The higher identification percentages for the distinction between 2 rows and 6 rows (CVP = 89.3%), between naked and hulled barley (CVP = 85.3%) and between sowing periods (between 65.6% and 77.4%) are promising for documenting the characteristics of archaeological barley. Indeed, barley is found charred in archaeological contexts, preventing the study of grains taxonomy using genetics. Further studies should include charring experiment of the modern diversity in order to build a reference collection of known characteristics (such as 2-row/6-row or naked/hulled types) directly comparable with archaeological charred grains. I would be then possible to explore the diachronic evolution of barley and the factors shaping its diversity over time.

## Supporting information

**S1 Table. Barley accessions information as varieties, subspecies, hulled or naked barley, sowing period, agronomic selection number of grains sampled per accession.**
(EPS)

**S1 Fig. Morphometric protocol applied to barley seeds.**
(EPS)

**S2 Fig. Rarefaction curves of barley seeds in X and percentage of population variation recorded in Y.**
(EPS)

## Acknowledgments

We gratefully thank the Biological Resource Center-INRAE Clermont Ferrand for providing the studied grains.

## Author Contributions

**Conceptualization:** Laurent Bouby, Vincent Bonhomme, Allowen Evin.

**Data curation:** Camille Dham.

**Formal analysis:** Angèle Jeanty, Allowen Evin.

**Funding acquisition:** Allowen Evin.

**Methodology:** Angèle Jeanty, Sarah Ivorra, Allowen Evin.

**Project administration:** Allowen Evin.

**Resources:** François Balfourier, Clément Debiton, Camille Dham.

**Software:** Allowen Evin.

**Supervision:** Laurent Bouby, Jérôme Ros, Allowen Evin.

**Validation:** Laurent Bouby, Sarah Ivorra, Jérôme Ros, Allowen Evin.

**Visualization:** Vincent Bonhomme, Sarah Ivorra, Allowen Evin.

**Writing – original draft:** Angèle Jeanty, Allowen Evin.

**Writing – review & editing:** Laurent Bouby, Vincent Bonhomme, François Balfourier, Clément Debiton, Sarah Ivorra, Jérôme Ros, Allowen Evin.

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
