## [Decision Letter · Decision Letter 0]

8 Jan 2023

PONE-D-22-27124Barley systematics and taxonomy foreseen by seed morphometric variationPLOS ONE

Dear Dr. Jeanty,

Thank you for submitting your manuscript to PLOS ONE. After careful consideration, we feel that it has merit but does not fully meet PLOS ONE’s publication criteria as it currently stands. Therefore, we invite you to submit a revised version of the manuscript that addresses the points raised during the review process.

We look forward to receiving your revised manuscript.

Kind regards,

Muhammad Abdul Rehman Rashid, PhD

Academic Editor

PLOS ONE

Journal Requirements:

“This work was supported by the European Research Council (ERC) under the European Union’s Horizon 2020 research and innovation programme (grant agreement No. 852573). We gratefully thank the Biological Resource Center-INRAE Clermont Ferrand for providing the studied grains.”

“This work was supported by the European Research Council (ERC) under the European Union’s Horizon 2020 research and innovation programme (grant agreement No. 852573).

https://erc.europa.eu/homepage

Additional Editor Comments :

Please clarify the problem statement in introduction section clearly and justify it in discussion section. Its needed to answer the reviewer comments before further process.

Reviewers' comments:

Reviewer's Responses to Questions

**Comments to the Author**

1. Is the manuscript technically sound, and do the data support the conclusions?

Reviewer #1: Yes

Reviewer #2: No

Reviewer #3: No

Reviewer #4: Yes

2. Has the statistical analysis been performed appropriately and rigorously? 

Reviewer #1: Yes

Reviewer #2: No

Reviewer #3: Yes

Reviewer #4: Yes

3. Have the authors made all data underlying the findings in their manuscript fully available?

Reviewer #1: Yes

Reviewer #2: Yes

Reviewer #3: Yes

Reviewer #4: Yes

4. Is the manuscript presented in an intelligible fashion and written in standard English?

Reviewer #1: Yes

Reviewer #2: Yes

Reviewer #3: Yes

Reviewer #4: Yes

5. Review Comments to the Author

Reviewer #1: The manuscript by Jeanty et al. brings a rather important contribution to the field of archaeobotany that will be of relevance for researchers in other disciplines, such as agronomy and plant physiology. The demonstration that barley types (2 row/6row, grain nakedness, growth habit) can be discriminated based on grain morphology is immensely relevant and exciting. It also dispels the fears that interannual or environmental variation might render comparisons based on grain morphology ineffective.

I therefore recommend its publication in PLoS One with only some suggestions for the authors’ consideration.

Line 20 – no need to have a comma between “variation” and “on the grains”.

Line 68 – In fact, several recent genomic studies have highlighted the polyphyletic and multiregional origin of barley domestication. See, for example,

Allaby, R. G. (2015). Barley domestication: the end of a central dogma?. Genome Biology, 16(1), 1-3.

Poets, A. M., Fang, Z., Clegg, M. T., & Morrell, P. L. (2015). Barley landraces are characterized by geographically heterogeneous genomic origins. Genome biology, 16(1), 1-11.

Pankin, A., Altmüller, J., Becker, C., & von Korff, M. (2018). Targeted resequencing reveals genomic signatures of barley domestication. New Phytologist, 218(3), 1247-1259.

Line 77 – There are other ancient DNA studies from barley grains that have been useful to study its origins and evolution. See, for example,

Palmer, S. A., Moore, J. D., Clapham, A. J., Rose, P., & Allaby, R. G. (2009). Archaeogenetic evidence of ancient Nubian barley evolution from six to two-row indicates local adaptation. PLoS One, 4(7), e6301.

Hagenblad, J., Morales, J., Leino, M. W., & Rodríguez-Rodríguez, A. C. (2017). Farmer fidelity in the Canary Islands revealed by ancient DNA from prehistoric seeds. Journal of Archaeological Science, 78, 78-87.

Line 112 – For the sake of readability, please make the numbering of aims here the same as in the Abstract.

Line 127 – Why 66 instead of 58 accessions? What is the difference between a “variety” and an “accession” in this context? Sounds confusing.

Line 158 – why only 784 SNPs when previously (line 60) there were 1056 SNPs? I presume this is the number of SNPs used for these particular accessions, but please clarify.

Line 158 – Also, as a suggestion, it would be nice to do a PCA for the accessions used, based on the genetic data, colouring the points with the same colour-scheme used in Figure 1 for 2-rowed hulled, 2-rowed naked, etc. It is not necessary for the overall story but it would be an interesting addition.

Also, please make the SNP data available as an Excel Spreadsheet as a supplementary file. This open-data practice is standard in the biological sciences.

Line 188 – Can we have a value for the “within”-variety variations or does the study described in Lines 138-142 make this unnecessary?

Line 234 – when discussing the results for genetic information and grain morphometrics, please address the issue that the number of SNPs used is to low for finding correlations between genetics and type. Intuitively, population structure will be more relevant than phenotype. For finding grain measurements causative SNPs a Genome-Wide Association Study (GWAS) study would have to be performed with hundreds of thousands of SNPs. You do mention in lines 357-359 studies looking at QTLs but perhaps refer to GWAS studies, for example:

Sharma, R., Draicchio, F., Bull, H., Herzig, P., Maurer, A., Pillen, K., ... & Flavell, A. J. (2018). Genome-wide association of yield traits in a nested association mapping population of barley reveals new gene diversity for future breeding. Journal of Experimental Botany, 69(16), 3811-3822.

Reviewer #2: In this paper, the authors had used a dataset of 58 varieties of French-origin barley to assess the taxonomic signal in grain measurements and test the impact of various factors on grain size and shape. They had also compared the relationship between morphometric and genetic proximity. I did not see any novelty in this manuscript. The work have not designed well. The methods of data collections and presentation of the results were not appropriate.

Please see my comments below:

- A small sample (58 French barley genotypes) size is a limitation of this study, as it may not provide enough power to detect differences or relationships that may exist.

- 54 barley genotypes grown in a field for only one year may not provide enough data to draw meaningful conclusions about their long-term performance or stability. If the goal of the study is to understand the characteristics and behavior of grain measurements, it may be necessary to grow them for multiple years or in different environments to get a more complete understanding of their characteristics.

Reviewer #3: The authors describe a new method to cluster barley seeds from their morphology e.g. into 2-row and 6-row types. This may be useful in many respects, and they claim that the method is relevant for analysis and classification of archaeological grains. However, this remains very speculative for a number of reasons: First of all, the analysis presented in this manuscript are all conducted on non-charred grains. This is contradictory to the idea, that the method is developed to be used for archeological grains, since the absolute majority of archaeological grains are charred. Other studies have shown that the morphology of grains changes when they are charred. We can therefore not be sure, if the method is applicable for charred grains. Secondly, the authors do not demonstrate an analysis of any archaeological grains and how well the method can cluster them into subgroups. This is essential for the conclusion that the method is relevant for archaeobotany. Another important discussion point is that (pre-historic) archaeological barley grains are significantly smaller than extant varieties. This suggests that extensive breeding has increased the size of barley grains over the last millennia. Therefore it is unclear if the method is applicable to pre-historic barley grains.

The method developed may be very useful in many respects. However, for the above reasons the conclusion drawn by the authors, that the method can be sued to analyze pre-historic grains is not documented by the presented data. Interpreted in another context the method could certainly be acceptable for publication.

Reviewer #4: This is a timely paper with potentially significant impacts on a number of fields, including archaeology, as the authors rightly suggested. The authors conducted vigorous analysis on a large dataset (58 varieties) of barley caryopses. The results suggest that caryopses metrics correspond to ear morphotypes (2/6-row; hulled/naked), as previously speculated by scholars but now approved with robust quantified evidence. The new data concerning sowing times and grain metric variations is excellent and constitute a significant contribution. I have a couple of referencing comments. The conclusion on that growing condition does not impose major effects on grain morphology, in relative terms, resonates with Ritchey and colleagues' work (2022), suggesting human choice and varietal difference played a key role in regional variations of barley grain morphotypes in prehistory, not the environment. Additional reference to Liu et al. 2017 on the eastern dispersal of barley will be helpful in covering a global perspective of early barley cultivation. Otherwise, this should be published, and I have no qualms with the manuscript.

Liu, X., D.L. Lister, Z. Zhao, C.A. Petrie, X. Zeng, P.J. Jones, R. Staff, A.K. Pokharia, J. Bates, R.N. Singh, S.A. Weber, G. Motuzaite Matuzeviviute, G. Dong, H. Li, H. Lü, H. Jiang, J. Wang, J. Ma, D. Tian, G. Jin, L. Zhou, X. Wu & M.K. Jones, 2017. Journey to the East: diverse routes and variable flowering times for wheat and barley en route to prehistoric China. PLOS ONE, 12(11), e0209518.

Ritchey, M. M., Y. Sun, G. Motuzaite Matuzeviciute, S. Shaoda, A. K. Pokharia, M. Spate, L. Tang, J. Song, H. Li, G. Dong, P. Vaiglova, M. Frachetti and X. Liu, 2022. The Wind that Shakes the Barley: the role of eastern Eurasian cuisines and environments on barley grain size. World Archaeology, 53(1): 1-18.

6. PLOS authors have the option to publish the peer review history of their article (what does this mean?). If published, this will include your full peer review and any attached files.

Reviewer #1: **Yes: **Hugo R. Oliveira

Reviewer #2: No

Reviewer #3: No

Reviewer #4: No

---

## [Author Response · Author response to Decision Letter 0]

27 Jan 2023

Institut des Sciences de l’Evolution Montpellier

Université Montpellier, CNRS, IRD, EPHE

2 Place Eugène Bataillon

34095 Montpellier Cedex 05

France

January 17th, 2023

PLOSONE

Dear Editor, 

Please find enclosed our reviewed manuscript entitled « Barley systematics and taxonomy foreseen by seed morphometric variation ».

We have agreed to most of the reviewers comments and made the following modifications:

Reviewers #1

1. Line 20 – no need to have a comma between “variation” and “on the grains”.

Comma has been removed. (Line 20).

2. Line 68 – In fact, several recent genomic studies have highlighted the polyphyletic and multiregional origin of barley domestication. See, for example,

Allaby, R. G. (2015). Barley domestication: the end of a central dogma?. Genome Biology, 16(1), 1-3.

Poets, A. M., Fang, Z., Clegg, M. T., & Morrell, P. L. (2015). Barley landraces are characterized by geographically heterogeneous genomic origins. Genome biology, 16(1), 1-11.

Pankin, A., Altmüller, J., Becker, C., & von Korff, M. (2018). Targeted resequencing reveals genomic signatures of barley domestication. New Phytologist, 218(3), 1247-1259.

Text reformulated and references added: “Barley domestication has long been thought to be monophyletic (Badr et al. 2000), but recent genetic analyses of current varieties support the hypothesis of a polyphyletic and multiregional origin of barley domestication (Morrell et al. 2007, Allaby 2015, Poets et al. 2015, Pankin et al. 2018)”. (Lines 74 - 77).

3. Line 77 – There are other ancient DNA studies from barley grains that have been useful to study its origins and evolution. See, for example,

Palmer, S. A., Moore, J. D., Clapham, A. J., Rose, P., & Allaby, R. G. (2009). Archaeogenetic evidence of ancient Nubian barley evolution from six to two-row indicates local adaptation. PLoS One, 4(7), e6301.

Hagenblad, J., Morales, J., Leino, M. W., & Rodríguez-Rodríguez, A. C. (2017). Farmer fidelity in the Canary Islands revealed by ancient DNA from prehistoric seeds. Journal of Archaeological Science, 78, 78-87.

References added: “Palmer et al. 2009, Hagenblad et al. 2017”. (Line 86).

4. Line 112 – For the sake of readability, please make the numbering of aims here the same as in the Abstract.

Done, modified in the abstract : “(1) to assess the taxonomic signal in barley grain measurements comparing 2-row and 6-row subspecies, and naked and hulled types; (2) to test the impact of the sowing period and interannual variation on the grains size and shape; (3) to investigate the existence of morphological differences between winter and spring types; and finally (4) to contrast the relationship between the morphometric and genetic proximity.” (Lines 18 – 22).

5. Line 127 – Why 66 instead of 58 accessions? What is the difference between a “variety” and an “accession” in this context? Sounds confusing.

Explanations have been added: “An accession includes grains from a single variety grown and sampled the same year in the same field. [..] These last 12 accessions (4 varieties x 3 years sown) varieties”. (Line 137 – 141)

6. Line 158 – why only 784 SNPs when previously (line 60) there were 1056 SNPs? I presume this is the number of SNPs used for these particular accessions, but please clarify.

Corrected “1056” with “784” in Line 65.

7. Line 158 – Also, as a suggestion, it would be nice to do a PCA for the accessions used, based on the genetic data, colouring the points with the same colour-scheme used in Figure 1 for 2-rowed hulled, 2-rowed naked, etc. It is not necessary for the overall story but it would be an interesting addition.

The SNP data are now available for reanalysis, however we do not think this addition will bring new information related to the questions and discussions raised specifically in the framework of this paper, we therefore decided not to include this analysis in our article.

8. Also, please make the SNP data available as an Excel Spreadsheet as a supplementary file. This open-data practice is standard in the biological sciences.

We provide all SNP data in a LabArchives repository (DOI provided in the Data reporting section).

9. Line 188 – Can we have a value for the “within”-variety variations or does the study described in Lines 138-142 make this unnecessary?

Since we did not analyse the morphometric variation at the scale of the variety or accession (except for the interannual diversity) we did not include all the morphometric descriptors of the grains, however all data are now available in a LabArchives repository (DOI provided in the Data reporting section).

10. Line 234 – when discussing the results for genetic information and grain morphometrics, please address the issue that the number of SNPs used is to low for finding correlations between genetics and type. Intuitively, population structure will be more relevant than phenotype. For finding grain measurements causative SNPs a Genome-Wide Association Study (GWAS) study would have to be performed with hundreds of thousands of SNPs. You do mention in lines 357-359 studies looking at QTLs but perhaps refer to GWAS studies, for example:

Sharma, R., Draicchio, F., Bull, H., Herzig, P., Maurer, A., Pillen, K., ... & Flavell, A. J. (2018). Genome-wide association of yield traits in a nested association mapping population of barley reveals new gene diversity for future breeding. Journal of Experimental Botany, 69(16), 3811-3822.

We reformulated the text and added references in the discussion section. The text now reads: “A possible explanation for the non-correlation between genetics and morphometrics may be the low number of SNPs used. To find grain measurements causative SNPs, a Genome-Wide Association Study (GWAS) would have to be performed with hundreds of thousands of SNPs (e.g. Sharma et al. 2018).”. (Lines 356 – 359)

Reviewers #2

1. A small sample (58 French barley genotypes) size is a limitation of this study, as it may not provide enough power to detect differences or relationships that may exist.

Of course our sample could be larger, but it is already the largest sample analysed through a geometric morphometric study focusing on cereal grains. We believe that it is powerful and fine enough to characterize the diversity of French modern barley. The other reviewers agree that this sampling is sufficient to answer the questions posed.

2. 54 barley genotypes grown in a field for only one year may not provide enough data to draw meaningful conclusions about their long-term performance or stability. If the goal of the study is to understand the characteristics and behavior of grain measurements, it may be necessary to grow them for multiple years or in different environments to get a more complete understanding of their characteristics.

The 58 barley genotypes serve as a basis for the morphological characterization of barley, the long-term follow-up should be carried out within the framework of another study which requires investigating the basics of current diversity. As already included in the manuscript, we had the opportunity to look at the interannual variability with 4 varieties that were grown for at least 2 years each. We agree with the reviewer in the sens that more can be done to understand better, but at least our results open new questions and preliminary results for interannual variability.

Reviewers #3

1. The authors describe a new method to cluster barley seeds from their morphology e.g. into 2-row and 6-row types. This may be useful in many respects, and they claim that the method is relevant for analysis and classification of archaeological grains. However, this remains very speculative for a number of reasons: First of all, the analysis presented in this manuscript are all conducted on non-charred grains. This is contradictory to the idea, that the method is developed to be used for archeological grains, since the absolute majority of archaeological grains are charred. Other studies have shown that the morphology of grains changes when they are charred. We can therefore not be sure, if the method is applicable for charred grains. Secondly, the authors do not demonstrate an analysis of any archaeological grains and how well the method can cluster them into subgroups. This is essential for the conclusion that the method is relevant for archaeobotany. Another important discussion point is that (pre-historic) archaeological barley grains are significantly smaller than extant varieties. This suggests that extensive breeding has increased the size of barley grains over the last millennia. Therefore it is unclear if the method is applicable to pre-historic barley grains. The method developed may be very useful in many respects. However, for the above reasons the conclusion drawn by the authors, that the method can be sued to analyze pre-historic grains is not documented by the presented data. Interpreted in another context the method could certainly be acceptable for publication.

The main goal behind this study is to apply geometric morphometrics to archaeological barley in order to characterize the past diversity of barley. This required first to develop a method to characterize the current diversity based on a large reference sample. To characterize the effects of charring is a significant objective for further archaeobotanical study and preliminary analysis are already available (Jeanty et al. in prep), however we preferred keep these analyses for a later publication. 

We modified the end of the conclusion according to the reviewer comment. This now reads as: “Further studies should include charring experiment of the modern diversity in order to build a reference collection of known characteristics (such as 2-row/6-row or naked/hulled types) directly comparable with archaeological charred grains. I would be then possible to explore the diachronic evolution of barley and the factors shaping its diversity over time”. (Lines 482 - 486)

Reviewers #4

2. This is a timely paper with potentially significant impacts on a number of fields, including archaeology, as the authors rightly suggested. The authors conducted vigorous analysis on a large dataset (58 varieties) of barley caryopses. The results suggest that caryopses metrics correspond to ear morphotypes (2/6-row; hulled/naked), as previously speculated by scholars but now approved with robust quantified evidence. The new data concerning sowing times and grain metric variations is excellent and constitute a significant contribution. I have a couple of referencing comments. The conclusion on that growing condition does not impose major effects on grain morphology, in relative terms, resonates with Ritchey and colleagues' work (2022), suggesting human choice and varietal difference played a key role in regional variations of barley grain morphotypes in prehistory, not the environment. Additional reference to Liu et al. 2017 on the eastern dispersal of barley will be helpful in covering a global perspective of early barley cultivation. Otherwise, this should be published, and I have no qualms with the manuscript.

Liu, X., D.L. Lister, Z. Zhao, C.A. Petrie, X. Zeng, P.J. Jones, R. Staff, A.K. Pokharia, J. Bates, R.N. Singh, S.A. Weber, G. Motuzaite Matuzeviviute, G. Dong, H. Li, H. Lü, H. Jiang, J. Wang, J. Ma, D. Tian, G. Jin, L. Zhou, X. Wu & M.K. Jones, 2017. Journey to the East: diverse routes and variable flowering times for wheat and barley en route to prehistoric China. PLOS ONE, 12(11), e0209518.

We added reference: “, Liu et al. 2017.” (Line 83).

Ritchey, M. M., Y. Sun, G. Motuzaite Matuzeviciute, S. Shaoda, A. K. Pokharia, M. Spate, L. Tang, J. Song, H. Li, G. Dong, P. Vaiglova, M. Frachetti and X. Liu, 2022. The Wind that Shakes the Barley: the role of eastern Eurasian cuisines and environments on barley grain size. World Archaeology, 53(1): 1-18.

We added reference: “In addition, it has been suggested that barley size was related to culinary systems and traditions in prehistoric Asia (Ritchey et al., 2022).” (Lines 115 – 116).

 Sincerely,

Mrs Angèle JEANTY and co-authors

---

## [Editor Report · Decision Letter 1]

18 Apr 2023

Barley systematics and taxonomy foreseen by seed morphometric variation

PONE-D-22-27124R1

Dear Dr. Jeanty,

We’re pleased to inform you that your manuscript has been judged scientifically suitable for publication and will be formally accepted for publication once it meets all outstanding technical requirements.

Kind regards,

Muhammad Abdul Rehman Rashid, PhD

Academic Editor

PLOS ONE
---

## [Editor Report · Acceptance letter]

24 Apr 2023

PONE-D-22-27124R1 

Barley systematics and taxonomy foreseen by seed morphometric variation 

Dear Dr. Jeanty:

I'm pleased to inform you that your manuscript has been deemed suitable for publication in PLOS ONE. Congratulations! Your manuscript is now with our production department. 

Kind regards, 

on behalf of

Dr. Muhammad Abdul Rehman Rashid 

Academic Editor

PLOS ONE